# Effect of Protein Conformation and AMP Protonation State on Fireflies’ Bioluminescent Emission

**DOI:** 10.3390/molecules24081565

**Published:** 2019-04-20

**Authors:** Cristina Garcia-Iriepa, Isabelle Navizet

**Affiliations:** Laboratoire Modélisation et Simulation Multi Échelle, Université Paris-Est, MSME UMR 8208 CNRS, UPEM, 5 bd Descartes, 77454 Marne-la-Vallée, France

**Keywords:** QM/MM, MD simulations, bioluminescence, emission spectra, fireflies

## Abstract

The emitted color in fireflies’ bioluminescent systems depends on the beetle species the system is extracted from and on different external factors (pH, temperature…) among others. Controlling the energy of the emitted light (i.e., color) is of crucial interest for the use of such bioluminescent systems. For instance, in the biomedical field, red emitted light is desirable because of its larger tissue penetration and lower energies. In order to investigate the influence of the protein environment and the AMP protonation state on the emitted color, the emission spectra of the phenolate-keto and phenolate-enol oxyluciferin forms have been simulated by means of MD simulations and QM/MM calculations, considering: two different protein conformations (with an open or closed C-terminal domain with respect to the N-terminal) and two protonation states of AMP. The results show that the emission spectra when considering the protein characterized by a closed conformation are blue-shifted compared to the open conformation. Moreover, the complete deprotonation of AMP phosphate group (AMP^2−^) can also lead to a blue-shift of the emission spectra but only when considering the closed protein conformation (open form is not sensitive to changes of AMP protonation state). These findings can be reasoned by the different interactions (hydrogen-bonds) found between oxyluciferin and the surrounding (protein, AMP and water molecules). This study gets partial insight into the possible origin of the emitted color modulation by changes of the pH or luciferase conformations.

## 1. Introduction

The bioluminescence of fireflies is a quite efficient natural process which consists of light emission produced by an enzymatic-chemical reaction. Apart from its outstanding efficiency, this process shows a high substrate specificity. For these reasons, fireflies’ bioluminescence has found numerous applications in the last decades [1,2,3,4,5,6,7,8]. Regarding the mechanism, the substrate (luciferin) inside the enzyme (luciferase) undergoes a multistep oxidative reaction in presence of molecular oxygen, adenosine triphosphate (ATP) and Mg^2+^ (Scheme 1). In particular, in a first step the luciferyl-adenylate intermediate is formed, catalyzed by the enzyme (reaction (1) in Scheme 1). Second, the high in energy intermediate (dioxetanone) is produced by reaction with the dioxygen molecule and release of adenosine monophosphate (AMP). Afterwards, decomposition of dioxetanone leads to the oxidation of luciferin producing the so-called oxyluciferin in the excited state (reaction (2) in Scheme 1). Finally, oxyluciferin decays to the ground state, emitting light (hν).

Although the mechanism is generally accepted, some crucial aspects such as the chemical nature of the light emitter or the emission color modulation observed when changing the pH, temperature or concentration of divalent metal cations still remain unsolved [9,10,11]. Moreover, it has been shown that the color emitted by different beetle luciferase species ranges from yellow-green, to orange or red though the substrate leading to the light emission is the same for all of them [12,13]. Hence, the different luciferase conformations and active sites should explain the different observed colors.

To get insight into the role of protein conformation on the color of the emitted light, several mutations of luciferase have been carried out, achieving in most cases a red-shift of the emission [14,15,16,17]. Focusing on the structure of the enzyme (luciferase) it consists of two domains, the N-terminal (400–500 amino acids) and the smaller C-terminal (110–130 amino acids), connected by a short flexible hinge region. These two domains play crucial role in the bioluminescent reaction as the active site is located at the interface between them, around the flexible linker. Luciferase falls within the superfamily of adenylating enzymes, for which two different conformations of the enzyme are adopted for catalyzing two subsequent reactions (1) and (2) in Scheme 1 [18,19,20]. In detail, this domain alternation catalytic mechanism lies in a ~140 degrees rotation of the C-terminal domain, once the first reaction is accomplished, to catalyze the second reaction. For luciferase, only biochemical evidences supported this domain alternation mechanism [21,22] until an engineered luciferase (PDB code 4G37) was produced to trap the enzyme in the second catalytic conformation with the C-terminal rotated (Figure 1), compared to the first catalytic conformation (PDB code 4G36) [23,24]. It was demonstrated that light was emitted when using the engineered luciferase by providing the luciferyl adenylate intermediate [23].

As aforementioned, changes in specific external factors also influence the color emission. A clear example is pH, as a yellow-green emission is observed at pH ~8 whereas a red emission is obtained at pH 6.5 [9]. In principle, the modification of pH could affect the chemical nature of oxyluciferin or the active site environment. Regarding the active site, different amino acids close to oxyluciferin could be sensitive to pH changes, as well as AMP which lies close to oxyluciferin once it is produced in the second catalytic reaction. Moreover, different chemical forms are possible for oxyluciferin, interconverted by tautomerization reactions or protonation/deprotonation of hydroxyl groups (phenol or enol groups) [25,26,27].

For all these reasons, in this study we propose the analysis of the influence of both the protein conformation and the protonation state of AMP on the emission (one of the possible factors sensitive to pH modifications). In particular, MD simulations have been performed to sample the relevant oxyluciferin-surrounding (water, protein, AMP) interactions. Then, the emission spectra have been simulated by computing the emission energies with quantum mechanics/molecular mechanics (QM/MM) methods. The phenolate-keto and phenolate-enol chemical forms of oxyluciferin have been selected for this study as they have been postulated to be the most probable light emitters in fireflies [28,29,30,31,32]. In addition, it has been shown that the emission mainly corresponds to that of the phenolate forms as an efficient excited state proton transfer (ESPT) between the phenol moiety and the surrounding can occur due to the high photoacidity of this hydroxyl group [26,33,34]. First, we analyze the effect of the protein conformation on the emission by considering the protein in an open or closed conformation (Figure 1). The C-terminal rotation could modify the active site nature, affecting the interactions with oxyluciferin and thus its emission. Furthermore, we investigate the effect of AMP protonation state on the emitted light as a partial study of the pH effect. At the biological pH, two protonation states of AMP could be possible (pKa 6.23), AMPH^−^ and AMP^2−^ (Figure 1). As AMP is quite close to oxyluciferin, its protonation/deprotonation could lead to different interactions and so emission energies. The role played by the protonation state of AMP on the emission has already been computationally investigated by using the fragment molecular orbital method [35]. Although the amino acids and water molecules within a radius of 7.5 Angstroms from oxyluciferin have been considered, dynamics to sample the different possible interactions have not been performed, being in some cases crucial to have a better picture of the system [26,36].

## 2. Results

In this section, we analyze the different interactions between the protein, AMP, water and oxyluciferin by performing MD simulations. In particular, two MD simulations (A and B) of 10 ns have been performed for each system, starting from the same initial conditions, to ensure an accurate sampling. Then, the influence on the emission spectra of the different interactions found along the simulations was checked by simulating the emission spectra as a convolution of gaussian functions of the emission energies computed with QM/MM methods for 200 snapshots (100 snapshots for each A and B simulations).

In particular, two sets of four systems (combination of two protein conformations and two protonation states of AMP) have been studied, one set for the phenolate-keto form of oxyluciferin and other one for the phenolate-enol form (Figure 1). These two oxyluciferin forms have been selected among the six possible chemical forms as they are the most probable forms leading to light emission [28,29,30,31,32]. To check the effect of the protein conformation on the emission spectra, the X-Ray structures of luciferase out of the Protein Data Bank corresponding to the first and second catalytic conformations (PDB codes 4G36 and 4G37 respectively) have been studied. As described in the introduction, the main structural difference between them is that the C-terminal domain of luciferase is rotated ~140 degrees in its second catalytic conformation (4G37), leading to a closed protein conformation (Figure 1). In addition, the protonation state of AMP could have a significant effect on oxyluciferin emission, as its protonation/deprotonation could form/disrupt hydrogen-bond interactions with oxyluciferin. For this reason, we have considered both the AMPH^−^ and the AMP^2−^ to build the system and simulate the emission spectra, so considering all these variables we have simulated the emission spectra for a total of eight systems: keto-4G36-AMPH, keto-4G36-AMP, keto-4G37-AMPH, keto-4G37-AMP, enol-4G36-AMPH, enol-4G36-AMP, enol-4G37-AMPH and enol-4G37-AMP.

The results section is divided in three parts. First, we examine the possible interactions between the oxygen of the phenolate moiety (for both keto and enol forms) of oxyluciferin with the protein (interactions with AMP are not feasible as it is far from this side of oxyluciferin). Second, we analyze the interactions between the oxygen atom of the keto group, for the phenolate-keto form of oxyluciferin, with AMP, water and the protein active site. Finally, we perform a similar study for the enol moiety of the phenolate-enol oxyluciferin.

### 2.1. Phenolate-Environment Interactions

We start by analyzing the interactions between the oxygen atom of the phenolate moiety (O1 in Figure 2) of oxyluciferin and the protein. In this case, the interaction between O1 and AMP is not possible as they are far from each other. Moreover, we study in this section the keto and enol forms of oxyluciferin, as both share this part of the chromophore in the two protein conformations (open and closed).

By analyzing the hydrogen-bond interactions between O1 and the protein, mainly two hydrogen-bond patterns have been found along the MD simulations for the eight systems under study. Pattern 1 is characterized by the interaction of O1 with 2 or 3 water molecules (Figure 2a) whereas no interaction with the protein is observed. While, for pattern 2, O1 interacts both with the side chain of ARG337 and with 1 or 2 water molecules (Figure 2b). As ARG337 is positively charged and close to O1, their interaction is straightforward, being stable during the simulation time once the hydrogen-bond is formed. In fact, it has been already shown the crucial role of this arginine in stabilizing a closed conformation of the protein and in creating a positively charged environment around the phenolate moiety of oxyluciferin [37].

As aforementioned, for each system we perform a set of two MD simulations (A and B) starting from the same initial conditions. In some cases, we find the two hydrogen-bond patterns for the same system. For instance, for keto-4G36-AMP, simulation A is characterized by the interaction of O1 with only water molecules (pattern 1), while during simulation B, ARG337 is closer to oxyluciferin and interacts with O1 (pattern 2). It is natural to wonder if these different hydrogen-bond patterns lead to distinct emission spectra. To answer this query, we have extracted snapshots from MD simulations A and B, each one characterized by one of the two hydrogen-bond patterns previously described. Then, their corresponding emission spectra have been simulated (see Methods section). By comparing these spectra, we can conclude that the different O1 hydrogen-bond patterns have no influence on the emission, as similar spectra have been obtained for the different sets of simulations (Figure 2c). In fact, the total number of hydrogen-bond interactions between O1 and the environment (either water or ARG337) is the same for both patterns and this could be the factor that governs the emission wavelength.

### 2.2. Keto-(Protein and AMP) Interactions

In the phenolate-keto chemical form of oxyluciferin, apart from the phenolate moiety it is also possible to have significant interactions between the oxygen atom of the keto group (O2 in Figure 3) and the surrounding. In this case, it is crucial to analyze not only the possible interactions between oxyluciferin and water or the protein, but also with AMP as it is quite close to this side of oxyluciferin (Figure 3).

First, we have analyzed the system considering AMPH^−^ (single protonated AMP, Figure 1). When oxyluciferin is placed inside the cavity of the protein in its open conformation, 4G36, we observe during the simulations hydrogen-bond interactions between the oxygen of the keto group (O2), one water molecule and AMPH^−^ (Figure 3a). However, when considering the closed protein conformation (4G37), only the interaction with AMPH^−^ is observed (Figure 3b). One possible reason to explain this finding is that less water molecules are inside the protein cavity for the closed protein conformation and so, no interaction between O2 and water is observed. To check this hypothesis, the number of water molecules along the simulation time within 3 Å of O2 is extracted. We can confirm that for the closed conformation (4G37) no water molecules are found close to O2 (only for a small fraction of snapshots one water is coming close) (black histogram in Figure 3c). However, for the open conformation one water molecule is close to O2 during more than half of the simulation time (and for some snapshots even two water molecules can be detected, red histogram in Figure 3c), leading to a large amount of hydrogen-bond interactions. Hence, we may suspect that the closing of the C-terminal domain could hamper the entrance of water molecules to the active site. To further check this assumption, we have also computed the number of water molecules around the phosphorus atom of the AMPH^−^ phosphate group and around the phenolate moiety (O1). For both, it is again observed that for the closed conformation (4G37) less water molecules are in the protein active site during the simulations (Appendix A).

Once identified the different interactions between O2 and the surrounding, inside the open and closed protein conformations, we investigate the influence of the different patterns found on the emission. After simulating the emission spectra for keto-4G36-AMPH and keto-4G37-AMPH we can conclude that the spectrum obtained when considering the closed conformation (4G37) is slightly blue-shifted (15 nm, 0.06 eV) compared to the one of the open conformation (Figure 3d), although the energy difference between them is quite small and within the method error. However, we can set that this trend can be due to the fact that O2 leads to more interactions with the environment (water and AMPH^−^) in the open conformation as the entrance of water is facilitated, resulting in a lower in energy emission. In fact, this larger amount of interactions in the thiazolone side could stabilize the LUMO orbital (more electron density located in the thiazolone ring compared to the HOMO, Appendix A), slightly decreasing the emission energy as observed in the simulated spectra. A similar reasoning has been previously followed focused on the interactions between the phenolate part and the environment to explain the emission shift [38].

Finally, we study the same systems but considering the completely deprotonated AMP (AMP^2−^ in Figure 1) to investigate what is the effect of removing the proton of the phosphate group on the hydrogen-bond patterns and so, on the emission. Starting with the open protein conformation, we observe that O2 interacts with one water molecule. Moreover, the conformation of the phosphate group is different from the one observed when doing the MD simulation considering AMPH^−^: the phosphate group points away from oxyluciferin (Figure 3a vs. Figure 4a). In detail, the phosphate group rotates leading to stable hydrogen-bond interactions with ARG437 (Figure 4a). However, a completely different scenario is found for the closed protein conformation. In this case, the phosphate conformation looks like the one of AMPH^−^ (Figure 3b vs. Figure 4b). Because ARG437 is farther away in the closed than in the open conformation, no interaction between ARG437 and AMP^2−^ phosphate group has been observed during the simulation. In fact, in the X-Ray starting structures, ARG437 is already farther from the phosphate group of AMP in the closed (4G37.pdb) than in the open conformation (4G36.pdb) (Appendix A). Hence, the phosphate group interacts with the closest residues (*e.g.,* THR343) and water molecules in the closed conformation. The water molecule that is H-bonding to O2 in the open conformation is interacting with AMP^2−^ in the close one (Figure 4b). For this reason, oxyluciferin moves inside the protein cavity searching alternative hydrogen-bond interactions, in particular with a NH group of AMP^2−^ and with the peptidic NH group of GLY316 (Figure 4b).

Thereafter, the emission spectra of keto-4G36-AMP and keto-4G37-AMP have been simulated to check the effect of the diverse hydrogen-bond patterns found for these systems. After simulation of the spectra, we can conclude that the emission when considering the closed conformation of the protein is significantly blue shifted (48 nm, 0.19 eV) compared to the open one (Figure 4c). So, the fact that oxyluciferin interacts in a different way with the protein cavity (it has moved inside the cavity) has a clear effect on the emission.

### 2.3. Enol-(Protein and AMP) Interactions

In this section we analyze the interactions between the hydroxyl group (i.e., OH) of the enol form of oxyluciferin (Figure 1) and the environment (both protein and AMP). In contrast to the keto form, in this case the hydrogen atom involved in the hydrogen-bond interactions belongs to oxyluciferin. Hence, it is expected that the hydrogen-bond patterns found along the simulations are different from the ones observed for the keto form of oxyluciferin, described in the previous section.

First, we analyze the interactions of the enol group for the two protein conformations considering AMPH^−^. For the open conformation, mainly two hydrogen-bond patterns have been found. Pattern 1 involves two H-bonds: O2-H of the enol group pointing towards the oxygen atom of the GLY341 peptide bond, and OH-phosphate group of AMPH^−^ interacting with O2 of the enol group (Figure 5a). Pattern 2 is characterized by the rotation of the enol group, which points towards AMPH^−^ (Figure 5b) leading to only one H-bond. For the closed protein conformation only one type of hydrogen-bond pattern has been found during the dynamic, corresponding to pattern 2 described for the open conformation: enol group pointing towards AMPH^−^ with O2-H interacting with the phosphate oxygen (Figure 5c). It has to be remarked that for pattern 1 of the open conformation (Figure 5a), the interaction between the enol group and the environment is larger than for pattern 2. Hence, a greater interaction between the enol group and the surrounding is observed for the open than for the closed conformation.

For this reason, we have checked the influence of this finding on the emission spectra. In the case of the phenolate-enol form, the emission of the system with the closed conformation is slightly blue-shifted (11 nm, 0.05 eV) (Figure 5d), following the same trend as for the keto form of oxyluciferin (Figure 3d). Although the energy difference found between the emission maxima is quite small, within the method error, the observed trend can be reasoned by the slightly larger number of interactions found between the enol group and the environment in the open conformation.

Finally, we have checked the influence of AMP deprotonation (AMP^2−^) on the hydrogen-bond interactions with phenolate-enol oxyluciferin and the environment. For both, the open and closed protein conformations, we observe during the MD simulation a strong hydrogen-bond interaction between the enol group of oxyluciferin and one oxygen atom of the phosphate group of AMP^2−^ (Figure 6a,b). Contrary to the keto oxyluciferin form, in this case the phosphate group keeps the same orientation for both protein conformations, (phosphate group pointing towards the enol group), due to its strong interaction with oxyluciferin. The main difference found between the two protein conformations is that more water molecules enter the protein cavity for the open conformation (Figure 6c), as already observed for keto-4G36-AMPH system. Hence, additional hydrogen-bond interactions between one water molecule, the enol group and AMP^2−^ are observed for the open conformation (Figure 6a). By simulating the emission spectra of these two systems we can conclude that again, the emission of the system with the closed protein conformation is blue-shifted (26 nm, 0.12 eV) due to the lower number of interactions between the environment and the enol group, as a consequence of the lowest amount of water molecules around it (Figure 6d).

## 3. Discussion

In the previous section we have presented the effect of the protein conformation and protonation state of AMP on the emission of phenolate-keto and phenolate-enol chemical forms of oxyluciferin, two possible light emitters of fireflies. First, we have analyzed the interactions of the phenolate moiety of oxyluciferin (keto and enol chemical forms) with the environment. Although a dynamic scenario of hydrogen-bond interactions has been found during the MD simulations (two different hydrogen-bond patterns involving water and/or ARG337, Figure 2), the same patterns have been found for the eight systems under study. In fact, the X-Ray structure of the part of the active site close to the phenolate moiety of both protein conformations, closed and open, are almost superimposed (Appendix A), in line with the similar interactions found during the MD simulations. In this case, same emission spectra have been simulated independently of the hydrogen-bond pattern, as both result in a similar stabilization of the HOMO orbital (Appendix A). Hence, the key factor responsible of the different simulated emission spectra must therefore lie in another part of the oxyluciferin molecule.

Motivated by finding the origin of the different simulated emission spectra, we have moved to the keto or enol side of oxyluciferin. Regarding the influence of the protein conformation, we can conclude that for all systems under study, the simulated emission spectra for the systems with the closed protein conformation (4G37) are blue-shifted compared to the ones with the open one (black vs. red lines in Figure 7a,b). However, this blue-shift is much more significant when considering AMP^2−^ (0.19 and 0.12 eV for phenolate-keto and enol respectively, dotted lines in Figure 7a,b) than when considering AMPH^−^ (0.06 and 0.05 eV for phenolate-keto and phenolate-enol respectively, solid lines in Figure 7a,b). In fact, the most blue-shifted simulated emission has been found for the 4G37-AMP systems (black dotted line in Figure 7a,b).

If we focus our attention on the effect of the protonation state of AMP on the emission, we can state that the system with the open protein conformation (4G36) is not sensitive to this modification, as similar emission spectra have been obtained for 4G36-AMPH and 4G36-AMP systems (red lines in Figure 7a,b). However, the systems with the closed protein conformation are more sensitive to the AMP protonation state, finding energy differences of the emission maxima of 0.13 and 0.09 eV for phenolate-keto and phenolate-enol respectively (black lines in Figure 7a,b). This result shows that if changing the pH alters the protonation state of AMP, the emission could be blue-shifted at high pH values (AMP deprotonated), leaving aside the influence of pH on the protein conformation or oxyluciferin chemical form, when considering the closed protein conformation, in line with the experimental results [9].

Moreover, by analyzing the relative intensity of the simulated emission spectra we can conclude that in general the emission of the enol form of oxyluciferin is significatively more intense than the one of the keto form (Figure 7). The less intense emission has been computed for the keto-4G37-AMP system. However, there is not a clear tendency between the emission intensity and the protein conformation or protonation state of AMP.

As discussed in the results section, the differences on the emission maxima can be mainly explained by the different hydrogen-bond patterns of oxyluciferin with the environment (both protein surrounding, AMP and water molecules). The most significative change has been observed for the keto-4G37-AMP system: as the phosphate group of AMP^2−^ points towards oxyluciferin, the side chains and water molecules close to the keto moiety are already interacting with the phosphate group and so, oxyluciferin moves inside the cavity searching other possible hydrogen-bond interactions. In fact, the largest emission energy shift has been found for this system, regarding the other phenolate-keto systems.

Thus, in this study we show that the protein conformation and the protonation state of AMP could influence the emission of fireflies’ bioluminescence. In particular, it has been observed that in general the emission spectra considering the closed conformation is blue-shifted regarding the open one. Moreover, it is interesting that when considering the open protein conformation, the simulated emission is not sensitive to the AMP protonation state, whereas for the closed protein conformation it is. Also, in some cases it has been observed that the open protein conformation facilitates the entrance of water in the protein cavity, leading to more hydrogen-bond interactions with oxyluciferin, red-shifting the emission.

## 4. Materials and Methods

### 4.1. Model Setup

In this study, eight different systems have been built by combination of different protein crystallographic structures (PDB 4G37 and 4G36), AMP protonation states (AMPH^−^ and AMP^2−^) and oxyluciferin chemical forms (phenolate-keto and phenolate-enol) (Figure 1). To build the system, the crystallographic structures 4G36.pdb and 4G37.pdb have been downloaded from the RSCB PDB website and the missing loops added with Disgro program [39]. Then, considering that the studied oxyluciferin forms are charged −1 and that the AMPH^−^ is also charged −1, some histidine residues have been protonated to neutralize the system using the H++ program [40], always keeping neutral the histidine residues close to the protein active site. In particular, for the 4G36 structure the 27, 46, 76, 419, 431 and 461 histidine residues and for the 4G37 the 76, 171, 310, 332, 419, 461 and 489 histidine residues were protonated (Appendix A). Moreover, when considering the deprotonated AMP (AMP^2−^) an additional histidine residue was protonated (the 171 and the 27 for the 4G36 and 4G37 structures, respectively). Afterwards, the substrate inside the crystallographic 4G36 and 4G37 structures, 5′-O-[N-(dehydroluciferyl)-sulfamoyl]adenosine (DLSA), has been replaced by the selected form of oxyluciferin and the AMPH^−^ or AMP^2−^. It should be noted that the 4G37 protein is a chemically engineered luciferase, designed to keep the second catalytic protein conformation (oxidative decarboxylation). In this regard, the groups and linkages used to keep this conformation have been deleted, as no accurate parameters are available to perform the MD simulations. Nevertheless, it has been checked that during the MD simulations of the 4G37 models, the protein conformation is kept closed although these linkage groups have been removed.

### 4.2. MD Simulations

MD simulations were performed with the Amber14 program [41]. The AMBERff14SB force field was selected to model the protein residues. The system built previously was solvated with an octahedral box of water molecules (TIP3P model [42]), ensuring a solvent shell of 10 Å around oxyluciferin. Then, the system was minimized and heated to 300 K. Finally, two production runs of 10 ns using the same starting point and time step (2 fs time step) but different seed were performed under NPT conditions (300 K and 1 atm), using periodic boundary conditions. Two production runs were performed to better sample the protein and oxyluciferin conformations. The Berendsen algorithm was selected to control the pressure and temperature [43]. Regarding the parameters (charges, bonds, angles and dihedral angles) of oxyluciferin, the hereafter procedure was followed: (1) parameters of the ground state optimized structure in the gas phase (B3LYP/6-311G(2d,p) have been selected as the starting point and a MD simulation was performed with this first parameters set (PS-GS1); (2) a random snapshot of this simulation was extracted and optimized in the ground state with QM/MM methods, retrieving a new parameters set (PS-GS2); (3) a MD simulation with the PS-GS2 set was performed; (4) a random snapshot of this simulation was extracted and optimized in the first singlet excited state with QM/MM methods, retrieving the parameters for the excited state (PS-ES); (5) two MD simulations starting from the same structure but different seed were performed using the PS-ES parameters and finally, (6) 100 snapshots were extracted from each simulation (200 snapshots in total each system) to compute the emission energies at the TD-DFT level using the B3LYP functional. The parameters for AMPH^−^ and AMP^2−^ have been already designed by our group [38,44,45].

### 4.3. QM/MM Calculations

QM/MM calculations have been performed using a QM/MM coupling scheme [46] between Gaussian09 [47] and Tinker [48]. The B3LYP functional and the 6-311G(2d,p) basis set were selected, as their suitability for this chromophore has been previously reported [25,26,49,50]. Nevertheless, a benchmark with the CAM-B3LYP and M062X functional has been done, showing similar trends (Appendix A). The oxyluciferin chromophore was treated at the QM level whereas the rest of the system was considered at the MM level. The interaction between the QM charge density (electrons and nuclei) and the external electrostatic potential of the MM part was computed by the electrostatic potential fitted (ESPF) method [46]. The microiterations technique [51] was used to converge the MM subsystem geometry for every QM minimization step. The emission spectra were simulated by a convolution (full-width at half-maximum of 0.2 eV) of gaussian functions for the emission energies computed at the TD-DFT level using the B3LYP functional for 200 snapshots extracted from the MD simulation (100 snapshots for each duplicated MD simulation done for every system) performed with the PS-ES parameters set.

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
