# Peer review of "Effect of Protein Conformation and AMP Protonation State on Fireflies’ Bioluminescent Emission"

_molecules, 2019, doi:10.3390/molecules24081565_

Round 1
Reviewer 1 Report
The work by Garcia-Iriepa et. al tries to investigate the influence of the protein environment and the pH on the emission spectra of the oxyluciferin by means of MD simulations and QM/MM calculations. The authors considered two different protein conformations and two protonation states of AMP for both enol and keto form oxyluciferin. This study show that the protein conformation and the pH values could influence the hydrogen-bond interactions and the protonation state of AMP so that could affect the emission of fireflies’ bioluminescence. The conclusions gave out a reasonable explanation on the multicolor of bioluminescent system. Thus I basically think that this work is potentially interesting to the general readership of the journal.
I have only one question about this article: the authors attributed the pH influence mainly on the different protonated states of AMP under different pH value. According to this article, the protein conformation and the forms of oxyluciferin (enol or keto) also strongly affects the emission spectrum. I wondering that whether or not the pH will influence the protein conformation and the forms of oxyluciferin. The authors could illustrate this in the paper.
Moreover, I thought the molecule orbitals of enol forms had better been put in the Supplementary Materials, too.
Author Response
Reviewer 1
The work by Garcia-Iriepa et. al tries to investigate the influence of the protein environment and the pH on the emission spectra of the oxyluciferin by means of MD simulations and QM/MM calculations. The authors considered two different protein conformations and two protonation states of AMP for both enol and keto form oxyluciferin. This study show that the protein conformation and the pH values could influence the hydrogen-bond interactions and the protonation state of AMP so that could affect the emission of fireflies’ bioluminescence. The conclusions gave out a reasonable explanation on the multicolor of bioluminescent system. Thus I basically think that this work is potentially interesting to the general readership of the journal.
I have only one question about this article: the authors attributed the pH influence mainly on the different protonated states of AMP under different pH value. According to this article, the protein conformation and the forms of oxyluciferin (enol or keto) also strongly affects the emission spectrum. I wondering that whether or not the pH will influence the protein conformation and the forms of oxyluciferin. The authors could illustrate this in the paper.
We agree with the referee that changes on the pH could lead to several modifications in the fireflies’ bioluminescent system such as the protein conformation, the chemical nature of the light emitter and the protonation state of AMP among others. In this work, a partial study of the effect of pH on the emission has been done by analyzing only the influence of the different protonation states of AMP. We have modified some sentences in the manuscript to clarify this point showing that we focus only on one factor related to the pH effect (AMP protonation/deprotonation).
Regarding the influence on the protein conformation, an ongoing project is being performed in collaboration with other theoretical group to do molecular dynamics at constant pH values and to analyze the influence on the emission spectra.
Considering the chemical form of oxyluciferin it has been already demonstrated that the emission comes mainly from a phenolate form, although not basic media is used (10.1021/ja017400m, 10.1021/ja3045212). For this reason, in this article we simulate the emission spectra of the most probable light emitters in fireflies, the phenolate-keto and phenolate-enol forms.
Moreover, I thought the molecule orbitals of enol forms had better been put in the Supplementary Materials, too.
As suggested by the referee, the molecular orbitals of the enol form have been added in Figure S2 in the Supplementary Materials.

Reviewer 2 Report
This manuscript studied the effect of protein conformation and AMP protonation on the bioluminescence of firefly via MD and QM/MM calculations. Please the authors consider below comments:
1. The article claimed that the emission spectra of oxyluciferin is simulated. Actually, only the vertical emission was calculated. Although the emission spectra were simulated by a convolution of gaussian functions, there is no more information than vertical emission. In figure 7, the emission intensity is the same for all spectra. What are the oscillator strength of each vertical emission? They should not be the same.
2. 4G36 is the adenylate-forming conformations of luciferase, which is the protein environment of the first catalytic reaction in Scheme 1. Right? In this reaction, oxyluciferin has not been produced. It is unreasonable to calculate emission energy of excited-state oxyluciferin in 4G36 conformation. If I am wrong, please correct me.
3. The Cartesian coordinates of QM/MM optimized geometries should be put in the supporting Information.
4. In the subsection of QM/MM calculations. The QM method should be TD B3LYP not B3LYP. I have to say B3LYP functional really has problem when handling H-bond systems and ESPT. At TD DFT computational level, the emission of 529,536 and 540 nm have no difference. Once different functional is employed, their order could change quietly possible, sometimes, even there is 50 nm difference among them.
Author Response
Reviewer 2
This manuscript studied the effect of protein conformation and AMP protonation on the bioluminescence of firefly via MD and QM/MM calculations. Please the authors consider below comments:
1. The article claimed that the emission spectra of oxyluciferin is simulated. Actually, only the vertical emission was calculated. Although the emission spectra were simulated by a convolution of gaussian functions, there is no more information than vertical emission. In figure 7, the emission intensity is the same for all spectra. What are the oscillator strength of each vertical emission? They should not be the same.
The simulated emission spectra have been built considering the emission energies and oscillator strength computed at the QM/MM level for 200 snapshots of each system under study. The spectra given in the manuscript have been normalized. We agree with the referee that the information of the oscillator strength could be interesting to analyze further differences between the systems.
For this aim, first the intensity of the emission maximum simulated for all the systems have been analyzed, finding that the largest one corresponds to phenolate-enol-4G37-AMP. This intensity has been taken as the reference to compute the relative intensity of the other spectra as shown in Figure 7. A sentence has been added in the manuscript to analyze the differences found between the relative intensities of the systems.
Moreover, as there is not a unique value of oscillator strength, 200 snapshots have been considered, a histogram with the oscillator strengths of the emission transition for all the systems under study is presented in Figures S4 and S5.
2. 4G36 is the adenylate-forming conformations of luciferase, which is the protein environment of the first catalytic reaction in Scheme 1. Right? In this reaction, oxyluciferin has not been produced. It is unreasonable to calculate emission energy of excited-state oxyluciferin in 4G36 conformation. If I am wrong, please correct me.
In fact it is still not clear which is the conformation of the protein when the light is emitted. Branchini et al. designed the 4G37 conformation aimed by the fact that luciferase belongs to the superfamily of adenylating enzymes, characterized by the domain alternation mechanism. However, the 4G37 is an engineered luciferase as a link between two amino acids should be included to keep the protein in that conformation. Not many experimental evidences support the domain alternation mechanism in fireflies. So, as the protein conformation during light emission is not clear, we have considered both the 4G36 and 4G37 for the emission spectra simulation.
Moreover, leaving aside the domain alternation mechanism, we consider the 4G36 and 4G37 structures as two models with quite different active site conformation with respect to oxyluciferin. As explained in the introduction, the 4G36 is open whereas the 4G37 is closed. Hence, we would like to analyze if this fact would affect the hydrogen-bond pattern around oxyluciferin (facilitating or not the water entrance) and so the emission energy.
3. The Cartesian coordinates of QM/MM optimized geometries should be put in the supporting Information.
As suggested by the referee we have included the geometries of the QM region for a representative snapshot (near the emission maximum) for the eight systems under study.
4. In the subsection of QM/MM calculations. The QM method should be TD B3LYP not B3LYP. I have to say B3LYP functional really has problem when handling H-bond systems and ESPT. At TD DFT computational level, the emission of 529,536 and 540 nm have no difference. Once different functional is employed, their order could change quietly possible, sometimes, even there is 50 nm difference among them.
We agree with the referee that the election of the DFT functional is crucial. For computing the emission of oxyluciferin we have chosen the B3LYP functional as has been previously shown its suitability for this system (see references in the computational methods of the manuscript). Regarding this work, we don’t study the ESPT process and no H-bonds are considered at the QM level as oxyluciferin leads to hydrogen-bond interactions with the protein active site, AMP or water molecules which are considered at the MM level.
Moreover, we agree that for some systems under study, the energy difference between the maximum emission spectra is quite small, being lower than 0.1 eV (within the TD DFT error). In order to clarify this point we have added some sentences along the manuscript.
Finally, the emission energy of one representative snapshot for each system (eight in total) have been computed using the CAM-B3LYP and M062X functional, following the same procedure used for the B3LYP one (Table S1). Similar trends have been observed although the absolute values are red shifted compared to the ones obtained with the B3LYP functional. In fact, the emission values computed with the B3LYP functional are the closest ones to the experiment.

Round 2
Reviewer 2 Report
The authors have replied the comments from reviewers and improved the article. Now it should be accepted. However, the language could be polished later by the editor. And, Table S1 has no units for the numbers.